# Effects of the Cardiac and Comorbid Conditions Heart Failure (3C-HF) Score at Admission on Life Space at Three Months after Hospital Discharge in Elderly Patients with Heart Failure: A Short Report

**DOI:** 10.3390/healthcare8040463

**Published:** 2020-11-05

**Authors:** Koichi Naito, Miyuki Kamo

**Affiliations:** 1Department of Physical Therapy, Hakuho College, Oji 6360011, Japan; 2Department of Cardiac Rehabilitation, Iwama Cardiovascular and Dental Clinic for Prevention and Care, Oji 6360002, Japan; 3Department of Rehabilitation, National Hospital Organization Kyushu Medical Center, Fukuoka 8108563, Japan; kamochan1221@gmail.com

**Keywords:** cardiac and comorbid conditions heart failure score, hemoglobin, life-space assessment, elderly, heart failure

## Abstract

We assessed 22 elderly patients with heart failure, 3 months after hospital discharge. We examined and compared factors that predict small life space after discharge with the same factors before discharge. Life-Space Assessment was used to classify the participants as either having “small living spaces” or “large living spaces”. We collected data regarding characteristics, Cardiac and Comorbid Conditions Heart Failure Score (3C-HF score), and evaluated motor function. We also collected data on the patients’ lifestyle habits. We investigated spreading life space at 3 months after discharge by mailing a questionnaire to the subjects. Using multiple logistic regression analysis, we were able to predict the patients having a small life space at 3 months after discharge by the Cardiac and Comorbid Conditions Heart Failure (3C-HF) score (odds ratio, 1.19; 95% confidence interval, 1.01–1.39; *p* = 0.038) at admission. Overall, the 3C-HF score at admission may be associated with the size of life space at 3 months after hospital discharge in elderly patients with heart failure. Future multicenter studies should be conducted to validate the results of this study by measuring post-discharge activity with a more objective index.

## 1. Introduction

With an aging rate of 28.1%, Japan has been categorized as a super-aging society. With a total population of 126.44 million people, some 35.6 million were aged 65 or older on 1 October 2018 [1]. Elderly patients with heart failure have been reported in aging societies, and the number of these patients increased by approximately 33% between 2008 and 2014 [2,3]. Therefore, the need for cardiac rehabilitation for heart failure patients is increasing. Seamless continuation of cardiac rehabilitation is important from hospitalization to outpatient visits; however, in Japan, the spread of outpatient cardiac rehabilitation has been particularly slow. The lack of staff and facilities has been reported as a reason [4], and immediate improvement is considered to be difficult. Therefore, physical activity at home after discharge is particularly important for patients with heart failure. The area of living space at home for patients with heart failure is thought to reflect the physical activity. A small living space can cause reduced motor function, a limited ability of patients to perform activities of daily living (ADL) [5], and an increased risk of frailty and mortality [6]. If it is possible to determine the effect of a small living space, post-discharge, during hospitalization, countermeasures may be taken. However, to date, factors that predict the small living space after discharge of patients with heart failure during hospitalization remains unestablished. This short report is the first of such attempts. Therefore, we aimed to examine the factors that predict small living space after discharge by comparing life space assessments before and after discharge.

## 2. Materials and Methods

### 2.1. Participants

The participants in this study were aged >65 years and were hospitalized for heart failure in the Kyushu Medical Center between November 2016 and December 2017. The participants exhibited no signs of dementia and were able to perform all ADL independently. Among those who were discharged to their homes, 22 patients (16 males, 6 females) consented to participate in this study and were selected. The study was conducted in accordance with the Helsinki Declaration. All participants provided written informed consent to participate. The Ethics Committee of the Kyoto Tachibana University approved the study (approval number 16–16).

### 2.2. Study Design

In this longitudinal study, we collected data regarding the characteristics of the patients including age, sex, height, weight, body mass index (BMI), educational history, brain natriuretic peptide levels (BNP), number of hospitalizations, and Cardiac and Comorbid Conditions Heart Failure (3C-HF) Score. We also evaluated the walking speed, grip strength, and timed up and go (TUG) test results of the patients. In addition, data on the lifestyle habits of the patients, such as living alone, smoking history, participation in community activities, and employment, were collected. We investigated the extent of life space at 3 months after discharge by mailing a questionnaire [7] to the subjects.

### 2.3. Walking Speed

To assess mobility, the walking speed of the patients was measured at discharge. The participants were asked to walk at their fastest speed on a 10-m walking course, which included a 3-m acceleration and 2-m deceleration zone at each end. The time was recorded for those who walked at least 5 m of the course. Walking speeds were calculated as m/s.

### 2.4. Grip Strength

To assess muscle strength, the grip strength of the patients was measured at discharge. The participants were asked to complete three trials of grip strength. Grip strengths of the right and left hands were measured in triplicate, using isometric dynamometry (T.K.K.5401 Grip-D; Takei Scientific Instruments Co., Ltd., Niigata, Japan). The highest value was selected from those measurements.

### 2.5. Timed Up and Go Test

To assess dynamic balance, the TUG of each patient was measured at discharge. The participants were asked to complete the TUG test at maximum speed. While performing the test, the participants were given verbal instructions to stand up from a chair, walk 3 m as quickly and safely as possible, cross a line marked on the floor, turn around, walk back, and sit down. The participants were given one practice trial to familiarize themselves with the task.

### 2.6. The Cardiac and Comorbid Conditions Heart Failure (3C-HF) Score

We calculated the 3C-HF score [8] using the data on admission. The 3C-HF score is a simple and valuable tool used in daily practice to improve the prognostic stratification of patients with heart failure. The 3C-HF score comprises cardiac and comorbid conditions such as the New York Heart Association class III–IV, left ventricular ejection fraction <20%, no beta-blockers, no renin–angiotensin system inhibitors, severe valve heart disease, atrial fibrillation, diabetes with micro or macroangiopathy, renal dysfunction, anemia, hypertension, and older age. After calculating these factors, the higher the score, the worse the prognosis.

### 2.7. Life-Space Assessment

We investigated the spread of life space at 3 months after discharge by mailing a questionnaire to the participants. Life-Space Assessment (LSA) is an index that evaluates the spatial extent of an individual’s life [7] by examining the individual’s normal living space during the month prior to the evaluation. Life space is defined as the distance one travels on a daily basis to carry out activities during a certain period. Therefore, the distance traveled by an individual at home is included, as well as surveys of frequency and degree of independence. Each level of life space is indicated by the distance from the individual’s bedroom and includes the individual’s distance travelled outside their home. The higher the total score, the wider the living space. According to previous research [9], a score of ≤56 points was classified as a “small life space” and ≥57 points as a “large life space”.

### 2.8. Statistical Analysis

Two groups, the “large life space” group and “small life space” group were defined. First, we performed Fisher’s exact test, Chi-square test, and *t*-test to compare the indexes (such as mean, standard deviation, and proportion) between the two groups. Next, we performed a logistic regression analysis to identify the factors that could predict having a small life space at 3 months after hospital discharge in elderly patients with heart failure. The entry probability for logistic analysis was set at a 0.05 level of significance. The model was simplified in a stepwise fashion by removing variables with a *p*-value > 0.05. The significance level was set at 5%, and IBM SPSS ver. 24.0 (IBM SPSS, Chicago, IL, USA) was used for all analyses.

## 3. Results

The characteristics of the participants and comparisons between the “large life space” and “small life space” groups are shown in Table 1. There were no significant differences in participant characteristics, motor function, or lifestyle habits between the groups. However, there were significant differences in the 3C-HF score (*p* = 0.01) and the hemoglobin levels (*p* = 0.02) in the domains of the 3C-HF score.

The sex, BMI, and 3C-HF scores recorded upon admission were subjected to logistic regression analysis using the risk of small life space as a dependent variable. The factor selected was the 3C-HF score upon admission. The Hosmer–Lemeshow statistic was χ^2^ = 7.51, adopting the null hypothesis. Based on the calculated odds ratio, the 3C-HF score upon admission was found to be significantly related to small life space (*p* = 0.038; Table 2).

## 4. Discussion

In our report, we surveyed 22 elderly patients with heart failure, 3 months after discharge from the hospital in order to examine factors that can predict the effect of small life space after discharge by using factors recorded before discharge. We found that the 3C-HF scores at admission may predict the spread of life space 3 months after discharge in these patients.

The percentage of patients with diabetes was 45.5% in the small space group as compared to 18.2% in large space group, although this difference was not statistically significant. Such disparities could be observed in many parameters, as indicated in Table 1. This could be due to the low number of subjects. In addition, the small number of participants may have contributed to the use of Fisher’s exact test, which was unlikely to determine a significant difference in the comparison of factors between the two groups.

In a qualitative study conducted in the United States, patients with chronic diseases were reported to be more prone to social isolation, including loss of friends and giving up on community activities and hobbies owing to their symptoms and treatment [10]. In addition, a qualitative study of Swedish patients with heart failure reported that the lack of mental and physical energy that such patients tend to feel can lead to limitations in performing social activities [11]. A similar result was observed in elderly patients with heart failure in our study. Furthermore, the severity of the disease was directly proportional to the degree of social isolation. The strong association between 3C-HF scores upon admission and small life space obtained in our study was considered to be caused by the above mechanisms. Furthermore, the association of 3C-HF scores with the prognosis in patients with heart failure could also be relevant to our findings [8].

It is reported that the 3C-HF score, based on easy-to-obtain cardiac and comorbid conditions, represents a simple and valuable tool to improve the prognostic stratification of heart failure patients in daily practice [8]. Although the 3C-HF score is useful, it is far from being sufficiently implemented. The reason for this may be the fact that it is a relatively new tool [8]. In addition, the fact that it has not been translated into Japanese may hinder its use in Japan.

The results of our short report show that the life space after hospital discharge might be small in patients with high 3C-HF scores upon admission. Previous studies have reported that a small life space is associated with decreased motor function, ADL disorders [5], increased health care utilization [12], and mortality [6]. Furthermore, it has been reported that a small life space appears before ADL disorders [7] and is a predictor of instrumental ADL [9]. Previous studies have also found that a variety of factors have been associated with living space size, including driving [13], social support [13], walking speed [13], age [14], and religious service attendance [14]. Therefore, it is necessary to take action before discharge from the hospital. By using the results of our study, we can select patients who are in particular need of coping with small life spaces before they are discharged from the hospital. In particular, the continuation of outpatient cardiac rehabilitation after discharge from the hospital and collaboration with the community can be implemented. We believe that participation and related preparations can be completed prior to discharge from the hospital. As it has also been reported that many patients who experienced hospitalization did not regain their former living space [15], it is important to implement strategies prior to discharge from the hospital.

Our study had some limitations. First, it was a single-center study and the number of cases was small, so our results may be limited. Therefore, it is still unknown whether our results will be implicitly observed in a wide population. Second, we used a questionnaire to examine details regarding life space after discharge, and we therefore may not have eliminated recall bias. Thirdly, because we were unable to conduct a baseline assessment of living space, we were not able to capture changes in living space. We plan to conduct multiple evaluations of living spaces and assess changes over time in a larger population.

## 5. Conclusions

Our short report shows that the 3C-HF score at admission may be associated with changes in life space at 3 months after hospital discharge in elderly patients with heart failure. Future multicenter studies are required to validate the results of this study by measuring post-discharge activity using an index with increased objectivity.

## Figures and Tables

**Table 1 healthcare-08-00463-t001:** Comparison between the “large life space” and “small life space” groups and characteristics of the patients.

Characteristics	“Large Life Space” (*n* = 11, 50.0%)	“Small Life Space” (*n* = 11, 50.0%)	*p*-Value
Age (years)	78.5 ± 7.3	76.7 ± 5.7	0.52
Female (%)	27.3	27.3	1
Height (m)	1.6 ± 0.1	1.6 ± 0.1	0.38
Weight (kg)	60.0 ± 16.1	60.2 ± 11.2	0.98
Body Mass Index (kg/m^2^)	23.4 ± 5.5	22.5 ± 3.2	0.67
Education (years)	12.0 ± 0.0	11.7 ± 0.9	0.34
Brain natriuretic peptide (pg/mL)	644.4 ± 403.3	648.3 ± 405.6	0.98
Number of hospitalizations (times)	2.2 ± 1.0	2.6 ± 0.8	0.25
Motor function			
Walking speed (m/s)	0.9 ± 0.3	1.0 ± 0.3	0.09
Grip strength (kg)	27.4 ± 10.3	22.9 ± 7.5	0.25
Timed up and go test (sec)	7.0 ± 2.7	8.8 ± 1.9	0.53
3C-HF score (points)	18.9 ± 8.2	29.6 ± 9.5	0.01
New York Heart Association class	0/18.2/45.5/36.4	0/0/54.5/45.5	0.33
(I (%)/II (%)/III (%)/IV (%))			
Left ventricular ejection fraction (%)	48.1 ± 13.8	38.6 ± 18.6	0.19
Atrial fibrillation (%)	36.4	54.5	0.67
Severe valve heart disease (%)	18.2	18.2	1
Complicated diabetes (%)	18.2	45.5	0.36
Creatinine (mg/dL)	1.1 ± 0.5	1.5 ± 0.7	0.11
Hemoglobin (g/dL)	13.8 ± 1.7	11.9 ± 1.8	0.02
Hypertension (%)	63.6	72.7	1
No beta-blockers (%)	9.1	27.3	0.59
No ACE-inhibitors/ARBs (%)	9.1	36.4	0.31
Lifestyle habits			
Living alone (%)	9.1	18.2	1
Current smoker (%)	9.1	0	1
Participation in community activities (%)	63.6	45.5	0.67
Employment (%)	45.5	27.3	0.66
LSA score	97.0 ± 19.3	32.1 ± 12.9	-

Values are mean ± standard deviation or proportion (%). 3C-HF score: Cardiac and Comorbid Conditions Heart Failure score; LSA: Life-Space Assessment; ACE: Angiotensin-converting-enzyme; ARBs: Angiotensin II Receptor Blocker.

**Table 2 healthcare-08-00463-t002:** Results of logistic regression analysis for risk of small life space (total 3C-HF score).

Predictor	Odds Ratio	95% Confidence Interval	*p*-Value
3C-HF Score (points)	1.19	1.01–1.39	0.038

Hosmer–Lemeshow test: χ^2^ = 7.51, *p* = 0.38 Model χ^2^ test: *p* = 0.005, Discrimination Rates: 63.6%.

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
