# Peer review of "Effects of the Cardiac and Comorbid Conditions Heart Failure (3C-HF) Score at Admission on Life Space at Three Months after Hospital Discharge in Elderly Patients with Heart Failure: A Short Report"

_healthcare, 2020, doi:10.3390/healthcare8040463_

Round 1

Reviewer 1 Report

Comments and changes suggested to the authors were taken into account.
Thanks

Author Response

Thank you for a very thorough peer review.

Reviewer 2 Report

After removing the hemoglobin data, the manuscript has very low impact, although now it is statistically and scientifically sounds which is of utmost importance. I do not have any other critical issues in this revised version.

Congrats!

Author Response

(The authors gave the same response as above.)

Reviewer 3 Report

In this small prospective study on post-discharge patients with heart failure, the authors aim to identify the characteristics of elderly patients (> 65 years) able to predict the “small life space” after discharge.

The study is well conducted and presents an interesting perspective to identify patients who are more fragile, with a greater tendency to social withdraw and on whom to act more actively to try to improve health status and prognosis.

Although presented clearly and linearly, the study is flawed by an extremely low number of patients enrolled. This is the main problem of the study that makes it not acceptable for publication in its current form. Given the high prevalence of heart failure in the general population (particularly in an elderly society such as Japan) and the low impact of the study procedures (non-invasive, easy to perform) the authors should aim to reach a much larger population to support their conclusions. This would also allow other predictive features of “small life space” to be identified.

Author Response

Thank you for your careful peer review.

As you pointed out, we believe that the small number of cases is a major problem.
For this reason, it is in the form of a short report.
We also added the phrase "Results may be limited" (Yellow highlight on line 168).
The word "short report" has been added to the "Conclusion" section (yellow highlight on line 175).

Round 2

Reviewer 3 Report

Dear authors, although you have changed the article into a "short report" I do not believe that the conclusions are adequately supported by the data. The sample is really too small.

Author Response

Thank you for the review. The sample size you pointed out is indeed small.
This point was raised by other reviewers and editors, and we withdrew it once,
but were instructed to resubmit it by other reviewers and editors,
as we could publish it if we changed it to a short report.
For this reason, We have resubmitted our short report with the addition
in the text that the results of this short report are limited.
We have also asked editage to correct the full text for English language
issues. We would appreciate it if you could take these circumstances into
account.